# Epigenetic profiling reveals key genes and cis-regulatory networks specific to human parathyroids

Youngsook Lucy Jung [1,11] ✉, Wenping Zhao[2], Ian Li [2], Dhawal Jain[1], Charles B. Epstein [3], Bradley E. Bernstein [3,4,5], Sareh Parangi[6], Richard Sherwood [7], Cassianne Robinson-Cohen[8], Yi-Hsiang Hsu[9,10], Peter J. Park [1] & Michael Mannstadt [2] ✉

In all terrestrial vertebrates, the parathyroid glands are critical regulators of calcium homeostasis and the sole source of parathyroid hormone (PTH). Hyperparathyroidism and hypoparathyroidism are clinically important disorders affecting multiple organs. However, our knowledge regarding regulatory mechanisms governing the parathyroids has remained limited. Here, we present the comprehensive maps of the chromatin landscape of the human parathyroid glands, identifying active regulatory elements and chromatin interactions. These data allow us to define regulatory circuits and previously unidentified genes that play crucial roles in parathyroid biology. We experimentally validate candidate parathyroid-specific enhancers and demonstrate their integration with GWAS SNPs for parathyroid-related diseases and traits. For instance, we observe reduced activity of a parathyroid-specific enhancer of the Calcium Sensing Receptor gene, which contains a risk allele associated with higher PTH levels compared to the wildtype allele. Our datasets provide a valuable resource for unraveling the mechanisms governing parathyroid gland regulation in health and disease.

The parathyroid glands are small endocrine organs that are essential for maintaining calcium homeostasis. They likely evolved from calcium-sensing cells within the gill buds of teleosts. This evolution provided terrestrial vertebrates, which are no longer immersed in a calcium-rich aquatic environment, with the ability to regulate calcium exchange[1]. In humans, there are four parathyroid glands located on the posterior surface of the thyroid gland in the neck. Their primary function is to respond to low serum calcium concentrations by

secreting parathyroid hormone (PTH), which acts directly on bones and kidneys to activate cellular processes that cause a net increase in serum calcium. Production and secretion of PTH appropriate to circulating concentrations of calcium is achieved through a negative feedback loop under the control of a transmembrane receptor protein, the calcium-sensing receptor (*CASR*), located on the surface of parathyroid cells[2]. Calcium homeostasis is primarily regulated by PTH and vitamin D. The parathyroid glands are the only source of PTH in the

[1]Department of Biomedical Informatics, Harvard Medical School, Boston, MA, USA. [2]Endocrine Unit, Massachusetts General Hospital and Harvard Medical School, Boston, MA, USA. [3]Broad Institute of MIT and Harvard, Cambridge, MA, USA. [4]Department of Cancer Biology, Dana-Farber Cancer Institute and Harvard Medical School, Boston, MA, USA. [5]Departments of Cell Biology and Pathology, Harvard Medical School, Boston, MA, USA. [6]Department of Surgery, Newton Wellesley Hospital and Massachusetts General Hospital, Harvard Medical School, Boston, MA, USA. [7]Division of Genetics, Department of Medicine, Brigham and Women's Hospital and Harvard Medical School, Cambridge, MA, USA. [8]Division of Nephrology, Department of Medicine, Vanderbilt University Medical Center, Nashville, TN, USA. [9]Hinda and Arthur Marcus Institute for Aging Research, Hebrew Senior Life, Boston, MA, USA. [10]Department of Medicine, Beth Israel Deaconess Medical Center and Harvard Medical School, Boston, MA, USA. [11]Present address: Division of Genetics and Genomics, Boston Children's Hospital, Boston, MA, USA. ✉e-mail: Youngsook_Jung@hms.harvard.edu; mannstadt@mgh.harvard.edu

body; any disruption in their function can lead to an imbalance in calcium homeostasis.

Although the role of PTH as a regulatory hormone has been well-established, less is known about the epigenetic mechanisms by which parathyroid cells regulate gene expression. In fact, in contrast with most other cell types, the epigenomic and transcriptomic landscape of parathyroid cells remains unexplored. Such information will be crucial for understanding basic cell physiology and disease processes. For example, the *CASR* gene expression is reduced in primary and secondary hyperparathyroidism, yet the underlying molecular mechanisms are not completely understood[2–4].

Recent advances in high-throughput sequencing-based assays have dramatically accelerated the investigation of tissue-specific gene regulation[5,6]. However, the study of human parathyroids has been hampered by the lack of a parathyroid cell line and the sparsity of human tissue samples. As a result, parathyroids have been absent from most of the consortium projects such as Roadmap Epigenomics[5], Genotype-Tissue Expression (GTEx)[7], and Human Cell Atlas[8].

Glial cells missing 2 (*GCM2*) is a transcription factor that is unique to the parathyroids and therefore provides an excellent focal point for unraveling parathyroid-specific transcription regulation. It is essential to the embryonic development of the parathyroid glands; in mice without *GCM2* the cells of the early parathyroid primordium fail to survive, and mature parathyroid glands do not form[9]. In humans, inactivating mutations of *GCM2* lead to congenital hypoparathyroidism, a condition characterized by insufficient PTH production and low calcium[10–13]. Robust expression of *GCM2* in the parathyroids is maintained throughout life; however, its role after development remains poorly understood. A recent study using a conditional knockout mouse model has shown that loss of *GCM2* induced in 8-week-old mice leads to a decrease in proliferation and an increase in apoptosis of parathyroid cells, suggesting *GCM2* plays a critical role in cell survival even after embryonic development[14].

Here, we produced a comprehensive map of the chromatin landscape of human parathyroid tissue. Integrating target genes of *GCM2*, we demonstrated that SNPs identified in GWAS of serum calcium and PTH concentrations in the population are enriched in parathyroid-specific regulatory regions. The relevance of these datasets is further exemplified by experimental data on a parathyroid-specific enhancer in the *CASR* gene, which contains an SNP identified by GWAS for serum PTH concentrations.

## Results
### Epigenomic and transcriptional profiles in parathyroids
We generated a set of epigenetic profiles from twelve human parathyroid adenoma samples using high-throughput sequencing-based assays. The datasets comprised (1) ChIP-seq data of the key parathyroid transcription factor, *GCM2*; (2) ChIP-seq data of active and repressive histone modifications including H3K4me3, H3K4me2, H3K4me1, H3K27ac, H3K36me3, H3K79me2, H2A.Z, H3K9ac, H3K27me3, and H3K9me3 along with *CTCF*; (3) ATAC-seq and DNase-seq data to determine chromatin accessibility; and (4) Hi-C data to identify higher-order chromatin interactions (Fig. 1a). We published a subset of the data (ChIP-seq data of histone modifications and *CTCF*) as part of the ENCODE project[6] but without parathyroid-specific analysis or functional assessments. We also generated bulk RNA-seq profiles collected from eight parathyroid adenoma samples. The t-SNE projection based on expression levels of parathyroid-specific genes revealed a tight clustering of the parathyroid samples. They were located near the pituitary gland and pancreas, two other endocrine organs. However, they were distinctly separated from the other tissue types profiled in the GTEx project, including endocrine organs such as the thyroid and adrenal glands (Fig. 1b, see the "Methods" section)[7].

To facilitate the functional interpretation of the chromatin datasets and the comparison with other tissue types, we generated chromatin state maps of the parathyroids using the histone modifications (Methods). Each chromatin "state" is defined by a combinatorial pattern of chromatin marks to enable robust annotation of genomic segments[15,16]. After producing chromatin state segmentation using the marks of H3K4me3, H3K4me1, H3K27ac, H3K36me3, H3K27me3, and H3K9me3 (which represent active promoters, enhancers, active enhancers, transcribed regions, Polycomb-regulated regions and heterochromatin regions, respectively), we compared the chromatin states in parathyroids with those in 98 other tissue or cell types from the Roadmap Epigenomics project[5]. Among the 17,829 active promoter states and 93,373 enhancer states in parathyroids, 8% of the promoters and 11% of the enhancers were parathyroid-specific (Supplementary Fig. 1a).

### *GCM2* targets active promoters and enhancer elements in parathyroids
Master transcription factors have been shown to form regulatory circuitries and hubs that are key to specific tissues. Therefore, we next focused on the transcription factor *GCM2*, an essential regulator of parathyroid cell survival and differentiation[10–14]. Among fully differentiated adult tissues, *GCM2* is expressed exclusively in the parathyroids[17]; however, the targeted genes and pathways of *GCM2* at a genome-wide scale are not known. *GCM2* ChIP-seq analysis on human parathyroid adenoma in biological replicates demonstrates that the vast majority (75.4%) of the 14,373 significant *GCM2* peaks were mapped to active promoters or regions proximal to active promoters or distal enhancers (19.8%, Fig. 1c, Supplementary Fig. 1b, and Supplementary Data 1). In contrast, *GCM2* binding was depleted in silent regions such as heterochromatin and Polycomb-regulated repressive regions (Fig. 1c). The histone modifications around GCM2 binding sites matched the patterns corresponding to active promoters or enhancers (Fig. 1d).

Importantly, a subset of *GCM2* binding sites at distal enhancer regions corresponded to broader and stronger regions of H3K27ac or H3K4me1, likely representing super-enhancers (clusters of multiple enhancers, Fig. 1d). As these regions and their associated genes are likely of critical importance to the tissue-specific transcriptional regulation and cell identity in parathyroid cells[18–20], we examined these super-enhancers bound by GCM2 in more detail. We defined super-enhancers based on the high signal intensities and region sizes of H3K27ac marks. Among 634 super-enhancer regions genome-wide defined by H3K27ac signals (Fig. 1e and Supplementary Data 2), 433 were targeted by GCM2, which was significantly higher relative to random chance ($p < 10^{-15}$, OR = 9.0, Fisher's exact test, Supplementary Fig. 1c), reflecting the importance of *GCM2* in driving the expression of parathyroid cell identity genes. The genes associated with these GCM2-bound super-enhancers included those previously identified as critical in parathyroid gland formation and function, such as *PTH, CASR, MAFB* (MAF BZIP transcription factor B) and *GATA3* (GATA binding protein 3) (Supplementary Fig. 1d)[21–24]. Notably, *GCM2* itself was also among these genes, consistent with studies of the related *gcm* in drosophila showing an autoregulatory circuit[25]. We also identified multiple genes with super-enhancers that were not previously recognized for their important roles in the parathyroids, such as *DNAH11* (Dynein Axonemal Heavy Chain 11), *PAX1* (Paired box protein 1), and *PTHLH* (parathyroid hormone-like hormone, *PTHrP*). These genes exhibited high expression levels in our RNA-seq dataset in the parathyroids (all within the top 5% of coding genes), further supporting their importance in parathyroid function, which will be discussed in more detail below. Genes containing super-enhancers were associated with gene ontology (GO) terms for RNA transcription regulation, metabolic process, and parathyroid development (Supplementary Data 3).

Next, we determined which super-enhancers are parathyroid-specific by comparing them to super-enhancers identified in 98 other cell/tissue types. We used the available H3K27ac ChIP-seq profiles from

the Roadmap Epigenomics project for this comparison. We identified 138 genes with super-enhancers that displayed specificity to parathyroids, appearing in no more than one additional tissue type (Supplementary Data 4). The list includes the core genes, *CASR*, *GCM2*, *PTH*, and *MAFB*. Collectively, the strong enrichment of *GCM2* binding sites at genes with super-enhancers underscores the distinct role of *GCM2* in the parathyroids.

## Dissecting parathyroid-specific cis-regulatory elements in the *CASR* and *PTH* genes

Through our genome-wide chromatin state analysis and comparison to the Epigenomics Roadmap data, we identified ~10,000 parathyroid-specific enhancers. To validate the usefulness of these maps, we focused on two genes essential for the function of parathyroid cells: *CASR* and *PTH*[26].

Although the *CASR* is most highly expressed in the parathyroids where it regulates PTH secretion, it is also expressed in many other tissues such as the kidneys, where it modulates calcium reabsorption and the pancreas, where it controls fluid secretion from pancreatic ducts[2]. The expression levels of the *CASR* differ between tissues[7,27] (Supplementary Fig. 2a). This variation is likely due to differences in enhancer utilization[28,29]. To uncover these cell-type-specific mechanisms, we examined DNase I hypersensitive (DHS) sites of the *CASR* in the tissues expressing the *CASR* (Fig. 2a and Supplementary Fig. 2b). There were striking differences in the usage of both promoter-proximal and distal enhancers of *CASR*. We identified prominent DHS peaks in the intronic region of the *CASR* specifically in the parathyroids. They were not found in the kidneys or pancreas. These parathyroid-specific DHS peaks were also bound by *GCM2* (Fig. 2a)

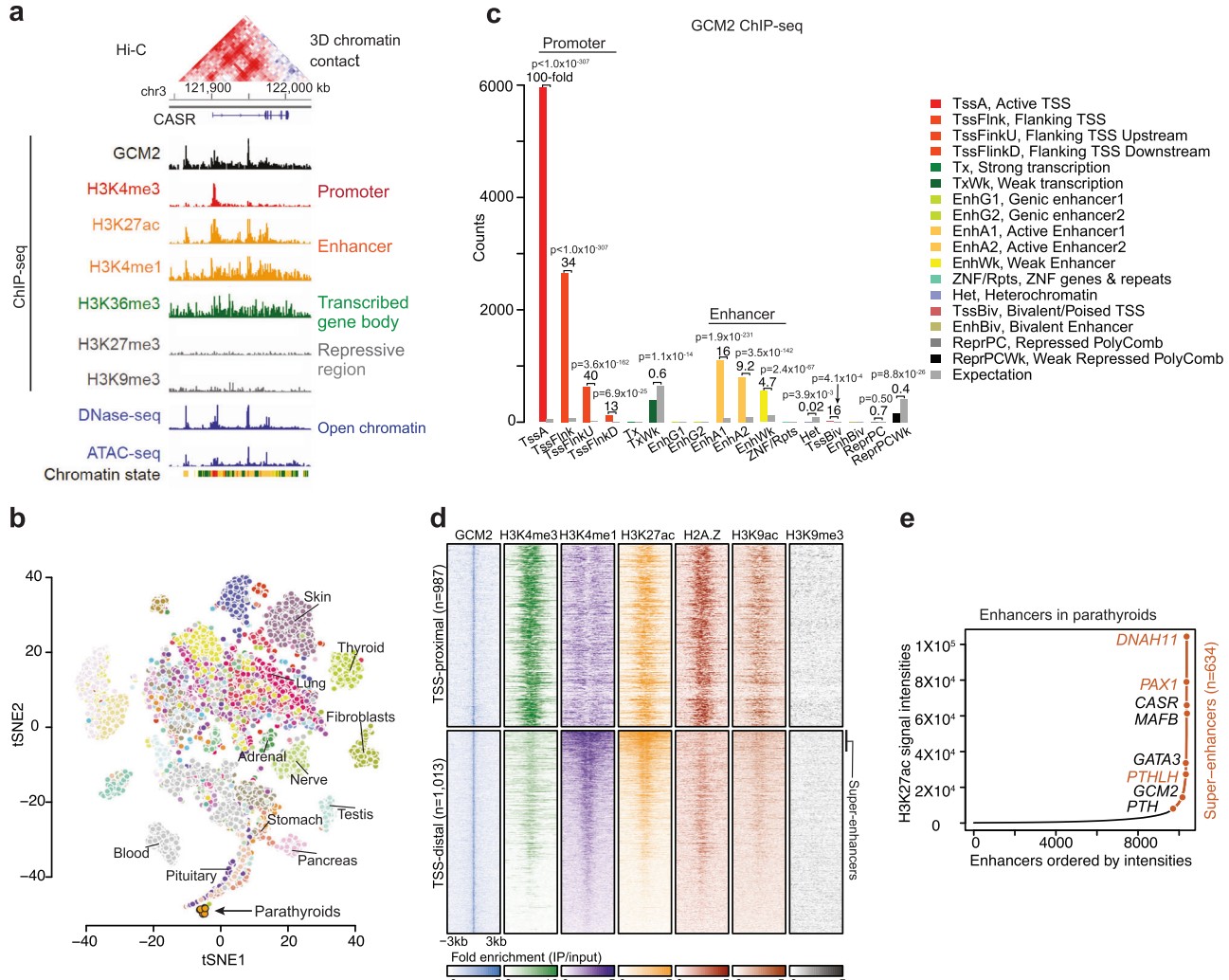

**Fig. 1 | Summary of datasets generated in this study and analysis of *GCM2* binding in parathyroids. a** List of select epigenomics profiles generated in this study and, as an example, the *CASR* gene. **b** t-SNE projection of RNA-seq from eight parathyroid adenoma samples (orange, arrow) based on parathyroid-specific expressions along with RNA-seq samples from 53 tissues profiled in the GTEx project where each dot represents each individual sample and each color represents each tissue type. **c** Distributions of *GCM2* binding mapped to chromatin states in parathyroids. The numbers positioned above the bars represent the ratios of observed numbers to expected numbers. *P*-values from a one-sided Fisher's exact test. TssA, Active TSS; TssFlnk, Flanking TSS; TssFinkU, Flanking TSS Upstream; TssFlinkD, Flanking TSS Downstream; Tx, Strong transcription; TxWk, Weak transcription; EnhG1, Genic enhancer1; EnhG2, Genic enhancer2; EnhA1, Active

Enhancer1; EnhA2, Active Enhancer2; EnhWk, Weak Enhancer; ZNF/Rpts, ZNF genes & repeats; Het, Heterochromatin; TssBiv, Bivalent/Poised TSS; EnhBiv, Bivalent Enhancer; ReprPC, Repressed PolyComb; ReprPCWk, Weak Repressed PolyComb. **d** Heatmap showing the patterns of histone modifications around *GCM2* peaks (±3 kb). Each row corresponds to each *GCM2* binding site. *GCM2* binding sites at the TSS-proximal (top) or distal to the TSS (bottom). **e** Super-enhancers in parathyroids. Enhancers sorted by H3K27ac signal intensities with super-enhancers defined by strong H3K27ac signals (orange line). Known key parathyroid genes are annotated in black and several genes previously not known to be key in adult parathyroids are annotated in orange; these genes were all found to be *GCM2* targets.

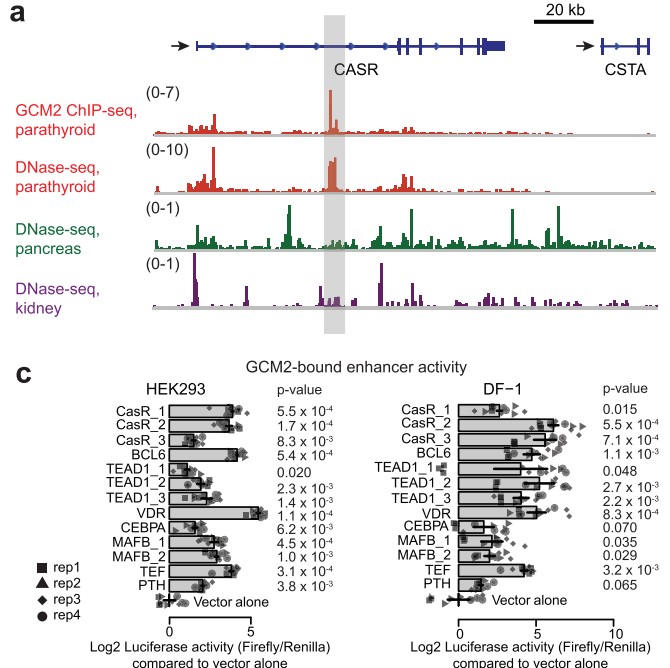

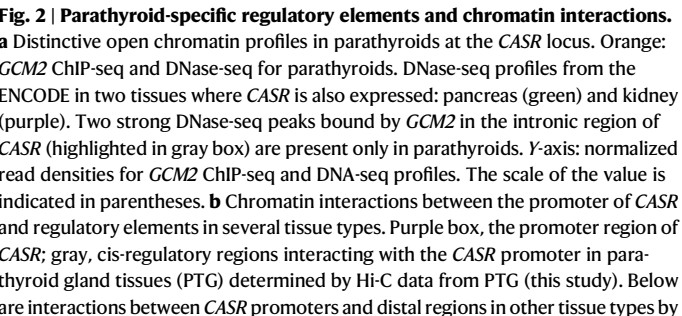

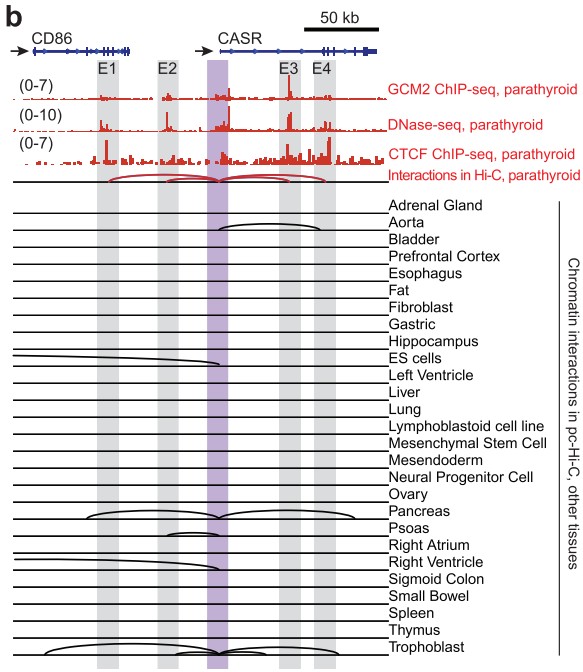

**Fig. 2 | Parathyroid-specific regulatory elements and chromatin interactions.** **a** Distinctive open chromatin profiles in parathyroids at the *CASR* locus. Orange: *GCM2* ChIP-seq and DNase-seq for parathyroids. DNase-seq profiles from the ENCODE in two tissues where *CASR* is also expressed: pancreas (green) and kidney (purple). Two strong DNase-seq peaks bound by *GCM2* in the intronic region of *CASR* (highlighted in gray box) are present only in parathyroids. *Y*-axis: normalized read densities for *GCM2* ChIP-seq and DNA-seq profiles. The scale of the value is indicated in parentheses. **b** Chromatin interactions between the promoter of *CASR* and regulatory elements in several tissue types. Purple box, the promoter region of *CASR*; gray, cis-regulatory regions interacting with the *CASR* promoter in parathyroid gland tissues (PTG) determined by Hi-C data from PTG (this study). Below are interactions between *CASR* promoters and distal regions in other tissue types by

promoter capture Hi-C. *Y*-axis for genomic profiles: normalized read densities for *GCM2* ChIP-seq, DNase-seq, and *CTCF* ChIP-seq. The scale of the value is indicated in parentheses. E enhancer element. **c** Parathyroid enhancer activity validations by luciferase assays in HEK293 and DF-1 cell lines. The enhancer elements were determined by H3K27ac, H3K4me1 and *GCM2* signals in parathyroids. *n* = 4 biologically independent samples; 3 repeated measurements for each biological sample. Rep biological replicate. Distinct dot shapes correspond to distinct samples. *P*-values were obtained through one-sided Student's *t*-tests on 4 biological replicates. The mean of 3 repeated measurements was used for each biological replicate. The bar graph displays the mean values of biological replicates with error bars of ±SEM (standard error of the mean).

suggesting that these enhancer elements in the intron of *CASR* may be fundamental to parathyroid function.

The activity of enhancers depends in part on three-dimensional enhancer-promoter interactions that can be determined by chromosome conformation capture-based sequencing[30,31]. To explore these chromatin interactions between promoters and distal enhancers, we generated high-resolution Hi-C maps of the parathyroids (Fig. 1a). We identified 22,434 significant interactions (FDR = 0.1, Supplementary Data 5, see the "Methods" section) between regulatory regions. Focusing on the *CASR* (Fig. 2b), we contrasted the interactions involving this gene in the parathyroids to those observed in other tissues using promoter-capture Hi-C assay (pcHi-C)[32]. In the parathyroids, we identified interactions between the promoter of *CASR* with four regulatory elements (E1–4) located in open chromatin and bound by *GCM2* and *CTCF*. E1 is located within *CD86*, a neighboring gene expressed in immune cells and silent in the parathyroids; E2 is in the intergenic region between the *CASR* and *CD86*. Given the lack of expression and the absence of open chromatin at the promoter of *CD86* in parathyroids, the first and second elements were likely enhancers of *CASR*, not *CD86*. E3 is the parathyroid-specific enhancer located in the intron of *CASR* as described above, and E4 is also located within the *CASR* gene. These four promoter-enhancer interactions at the *CASR* were specific to the parathyroids. Other tissues that express the *CASR*, such as embryonic stem cells and pancreas, exhibited chromatin loops at the *CASR* promoter that all interacted with different genomic regions (Fig. 2b).

Next, we explored parathyroid-specific enhancers in the *PTH* gene. In addition to a previously known enhancer ~5 kb upstream of the gene body[33], we identified a second enhancer ~20 kb upstream. This distal enhancer is parathyroid-specific, binds *GCM2*, has an open chromatin structure, and loops to the promoter of the *PTH* gene (Supplementary Fig. 2c).

We experimentally validated the activity of nearly a dozen of the identified enhancers using luciferase assays and a heterologous cell culture system (Fig. 2c). We selected enhancers in core parathyroid genes such as *CASR, PTH*, and *MAFB* that were bound by *GCM2* (Supplementary Data 6). Indeed, all of the putative enhancers exhibited significantly increased luciferase activity to varying degrees compared to empty vector controls (Fig. 2c, *p*-values by one-sided Student's *t*-test) using either HEK293 (human embryonic kidney) or DF-1 (chicken fibroblast) cells.

Collectively, these examples demonstrate the utility of our maps detailing parathyroid-specific open chromatin structures, enhancers, and three-dimensional chromatin interactions in understanding key factors for the identity and function of the parathyroid glands.

### Delineating a core regulatory transcription network in parathyroids

Identifying core regulatory circuits is critical for understanding cellular function. Our initial step involved the identification of TF binding sites that are enriched within open chromatin regions in parathyroid tissue. By conducting motif analysis, we identified a total of 26 TFs

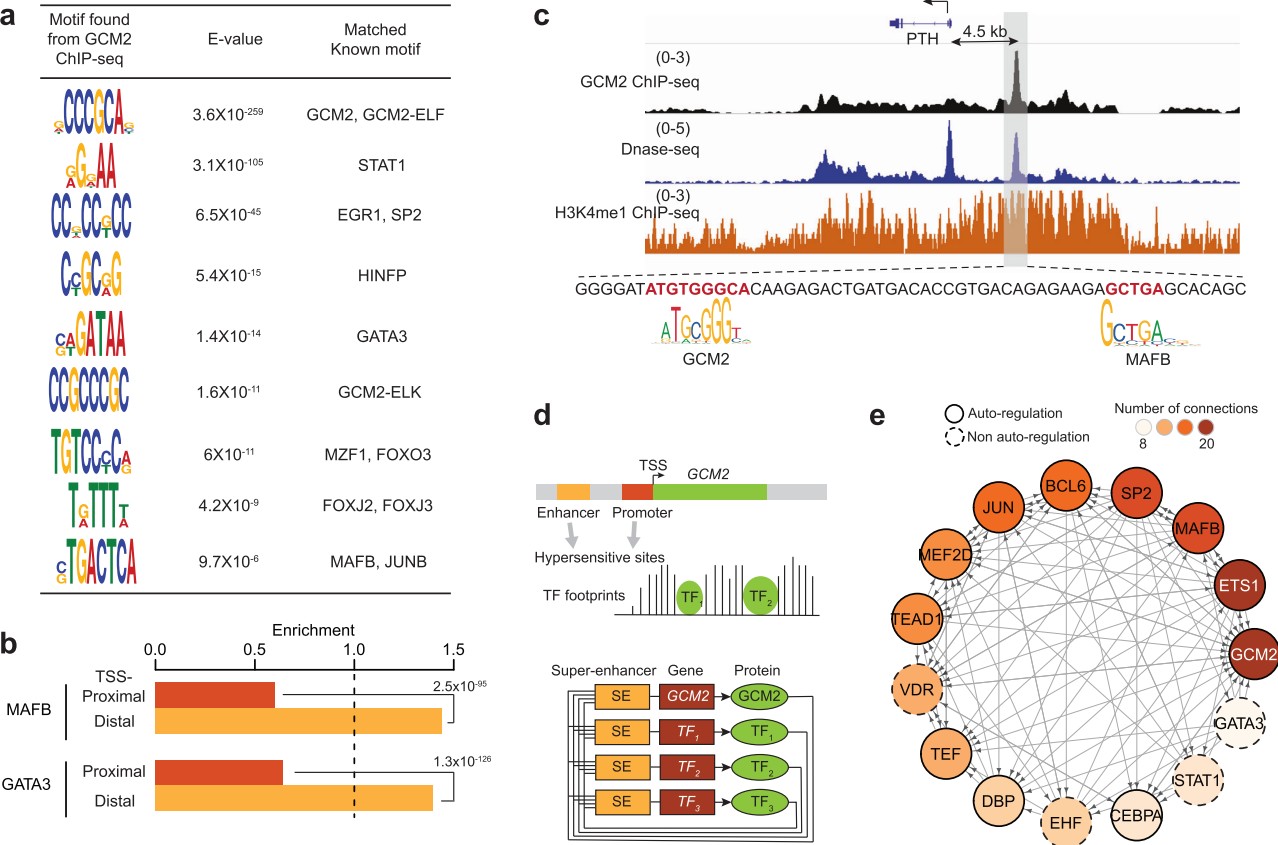

**Fig. 3 | Core cis-regulation circuit in parathyroids. a** Motif sequences from *GCM2* peaks and matched known TF motifs. **b** Binding prediction enrichment in TSS-proximal regions (red) and TSS-distal regions (orange) for *MAFB* and *GATA3*. *P*-values by a one-sided Fisher's exact test. **c** Predicted binding sites of *GCM2* and *MAFB* at the distal enhancer (4.5 kb from TSS) of the *PTH* gene. *GCM2* and *MAFB* are predicted to co-bind their core motif sites that are 40-bp apart. *Y*-axis: normalized read densities for *GCM2* ChIP-seq (black), DNase-seq (blue), and H3K4me1 ChIP-seq (orange). The scale of the value is indicated in parentheses. **d** Schematic illustration of identifying key TF regulators working with *GCM2*. We analyzed TF footprints in open chromatin regions of the *GCM2* gene (upper) and identified a set of TFs that co-regulate genes along with *GCM2* (lower). **e** A core TF regulation circuit in parathyroids. TFs with black solid circles are auto-regulated. The red color intensity indicates the number of connections (in and out combined).

(Supplementary Data 7; see the "Methods" section) with *GCM2* ranking among the top 4. Due to the unique expression of *GCM2* in parathyroids and the high prevalence of its motif in DHSs, we further explored the regulatory network among these TFs by leveraging *GCM2* ChIP-seq data. This allowed us to uncover a network of TFs centered around *GCM2*.

Focusing specifically on *GCM2*, we first investigated potential partnerships with other TFs by performing de novo DNA motif analysis on the *GCM2* ChIP-seq-binding sites. As expected, the discovered motif sequences from the *GCM2* peaks best matched the known *GCM2* motif CCCGCA, previously determined in in vitro assays[34]. Importantly, this analysis also revealed the enrichment of two heterodimer motifs, *GCM2-ELF* and *GCM2-ELK*. These heterodimer motifs were present in about one-third of all *GCM2* binding sites, suggesting that GCM2 frequently forms heterodimers with the ETS family transcription factors *ELF* or *ELK* (Fig. 3a)[35]. Our analysis also demonstrated frequent co-localization of *GATA3* and *MAFB* motifs with *GCM2* binding sites (Fig. 3a). Intriguingly, the co-localization of *GATA3* and *MAFB* motifs at *GCM2* binding sites was more commonly observed at distal enhancers than at promoters (Fig. 3b, *p*-values by one-sided Fisher's exact test), suggesting a role in the tissue-specific regulation of target genes. In Fig. 3c, we illustrate such co-localization at the *PTH* gene. The distal enhancer ~4.5 kb upstream of the *PTH* promoter contains a *GCM2* binding site with an *MAFB* motif close to the *GCM2* motif, consistent

with a previous report[24]. Other TFs such as *STAT1* (signal transducer and activator of transcription 1), *SP2* (specificity protein 2), *HINFP* (histone H4 transcription factor) and FOX (forkhead box) family members were also predicted to frequently co-localize with *GCM2* (Fig. 3a)[23].

Next, we defined a core regulatory network of TFs in parathyroids centered on *GCM2*. We hypothesized that members of such circuitry, including *GCM2*, bind to promoters or enhancers of other critical TFs associated with super-enhancers[20]. We first identified TF binding motifs at the promoter and enhancers of *GCM2*. We filtered for those TFs that are expressed in the parathyroids and whose genes themselves are associated with super-enhancers (Fig. 3d, see the "Methods" section). The regulatory circuit of TFs that we constructed revealed several important characteristics (Fig. 3e, Supplementary Fig. 3a and b): (1) the members in the core TF network largely overlapped with co-binding TFs predicted at *GCM2* (Fig. 3a), supporting an important role in regulating parathyroid genes, (2) a large fraction of the TFs in the circuit (11 out of 15) had auto-regulation properties (i.e., the protein binds to its enhancer for transcriptional autoregulation), analogous to similar circuitries in other cell types[20] (Fig. 3d lower), (3) the network included known key parathyroid TFs such as *MAFB*, *GATA3* and *VDR* (vitamin D receptor)[36], and (4) *GATA3* stands out as it targets other core TFs, but it is itself not targeted by other TFs except for *SP2*. This suggests that the pioneer TF *GATA3* likely sits at or near the top of this

regulatory network. This is consistent with the important role of *GATA3* in the formation of parathyroid glands[22] and experimental evidence showing that *GATA3* binds to a functional double-GATA motif within the *GCM2* promoter[21].

## Potential key parathyroid cell identity genes

After identifying key transcription factors, we generated a list of key genes previously not known to play important roles in parathyroid identity and function. To do this, we integrated cis-regulatory elements, expression data, and *GCM2* ChIP-seq profile using the following criteria—the genes (1) are associated with parathyroid-specific enhancers or super-enhancers, (2) are highly and specifically expressed in parathyroids (z-scores across tissues > 7) and (3) are targets of *GCM2* (Fig. 4a; see the "Methods" section). To identify genes expressed specifically in parathyroids, we compared the transcription levels of genes in parathyroids with those in 54 tissues from the GTEx (Supplementary Data 8). The generated list contains 78 genes which include known key genes such as *GCM2*, *PTH*, and *CASR*, but also those not previously known to fit this role such as *PAX1*, *PTHLH*, *DNAH11* (discussed below), *MDRT2* (Doublesex and mab-3-related transcription factor 2) and *TPM3* (Tropomyosin 3).

*PAX1*, a member of the paired box (PAX) family of TFs, was previously suggested to play a role in parathyroid development[37]. However, the role of *PAX1* in the adult parathyroid gland tissue has not been appreciated. The entire *PAX1* gene was broadly and strongly enriched for *GCM2* binding (Poisson distribution p value < $10^{-5}$) and contained parathyroid-specific super-enhancers, which contrasts with the repressive or poised chromatin states of this gene in other tissue types (Fig. 4b, upper), consistent with its unique and striking expression in the parathyroids (Fig. 4b, lower).

*PTHLH* (Parathyroid hormone-related peptide or *PTHrP*) has sufficient similarities to *PTH* to bind to the same *PTH/PTHrP* receptor. The *PTHrP* gene is expressed in many tissues and plays paracrine/autocrine roles[38], but to our knowledge, its function in parathyroids has not been studied. The *PTHLH* gene revealed high intensities of broad GCM2 binding and showed hallmarks of a super-enhancer in the parathyroids (Fig. 4c, upper). This is in sharp contrast to other tissues that express *PTHLH* but show much lower enhancer activity, not satisfying the super-enhancer criteria by H3K27ac signal intensity. The super-enhancer activity is likely the underlying mechanism for the high expression level of *PTHLH* in the parathyroids (Fig. 4c, lower).

*DNAH11* is a ciliary protein associated with primary ciliary dyskinesia. It ranked on the top among the genes in parathyroid by H3K27ac intensity (Fig. 1e). Although *DNAH11* was expressed in many other tissues, the transcription and enhancer activity of this gene in the parathyroids was the highest among the tissues and displayed multiple binding sites of *GCM2* (Fig. 4d).

These and other genes, including *MDRT2* and *TPM3* (Supplementary Fig. 4), exhibit higher levels of expression, enhancer activity, and specificity to the parathyroid gland, indicating important roles in the parathyroid regulation. Validating their function in vivo will require the generation of inducible knockouts for these genes in the parathyroid glands of mice.

## GWAS SNPs in parathyroid-specific enhancers

Genome-wide association studies (GWAS) have reported thousands of single-nucleotide polymorphisms (SNPs) associated with human disease or traits. Most of these SNPs are located in non-coding regions, with enrichment in enhancer elements[39–41]. Elucidating the molecular mechanisms underlying the genetic variants is challenging but profiling the epigenome of target tissues has been a fruitful approach to annotate GWAS hits in order to identify the biological basis of the association. However, the currently available chromatin maps miss many disease-relevant tissues. For example, determining the underlying genetic basis of GWAS SNPs for serum PTH, calcium levels, bone

mineral density or osteoporosis would benefit from having chromatin-state annotations of parathyroid tissue, as PTH secretion and the responsiveness of the parathyroids to extracellular calcium may be associated with these traits and diseases.

Using our dataset, we investigated GWAS SNPs associated with circulating PTH concentrations, even considering less stringent p-value thresholds of higher than $5 \times 10^{-8}$[42]. We found that a substantial fraction of the SNPs (18% among 2359 SNPs) was located and significantly enriched in the functional regions that we identified in parathyroids, such as active promoters, enhancers or super-enhancers (Fig. 5a, p-values by Fisher's exact test). Interestingly, 99 SNPs (23.7% of those in regulatory regions) were present within super-enhancers in parathyroids, which was significantly higher compared to genome-wide expectations (p < 0.001 by a permutation test; Supplementary Fig. 5a, see the "Methods" section). SNPs linked to bone density and osteoporosis displayed similar patterns (Fig. 5a). As negative control, SNPs linked to phenotypes unrelated to parathyroid function, such as facial morphology and schizophrenia, were not enriched in parathyroid enhancers, super-enhancers, or DHSs. Instead, they were predominantly enriched in weak Polycomb repressive regions (Supplementary Fig. 5b).

To begin unraveling the molecular mechanisms underlying GWAS SNPs associated with PTH concentrations, we focused on those SNPs located in proximity to *CASR*, a gene prominently involved in the regulation of PTH secretion (see preceding section). Our analysis revealed that all 7 of the GWAS-identified *CASR* SNPs were located in parathyroid enhancer regions. Specifically, rs9811123 was positioned at the parathyroid-specific DHS site bound by *GCM2* (Fig. 5b; see also Fig. 2, and Supplementary Data 9). SNPs in regulatory regions can result in altered binding affinity for TFs, which in turn can lead to changes in transcript abundance or alterations of mRNA splicing and microRNA binding. We hypothesized the SNP would functionally affect the *CASR* gene regulation by the TF binding affinity change. Indeed, when we examined GCM2 ChIP-seq reads covering this SNP, we observed allelic imbalance in SNP rs9811123 from one of our GCM2 ChIP-seq samples that was heterozygous at this position (Fig. 5c). Among 88 reads supporting this SNP site, 67% mapped to the major G allele while 33% reads mapped to the minor A allele, suggesting that this allele can affect enhancer activity likely through altered TF binding. Using motif analysis, the DNA sequence change at rs9811123 was predicted to lead to the disruption of binding of the parathyroid-expressed TF HINFP (Fig. 5d). HINFP is one of the TFs whose motif was frequently found at *GCM2* peaks (Fig. 3a). In addition, the decreased enhancer activity by this SNP was predicted using a support vector machine (SVM) model (see the "Methods" section)[43]. We validated the SNP effect using luciferase assays in two different cell lines (HEK293 and DF-1 cells, both endogenously expressing *HINFP*), showing significantly reduced enhancer activity with the minor allele at the rs9811123 region (p-values by one-sided Student's t-test, Fig. 5e).

Primary changes in PTH secretion result in changes in serum calcium. Having established a link between *CASR* SNPs and PTH levels, we asked if SNP rs9811123 is also associated with serum calcium levels. This SNP indeed also exhibited a significant association with serum calcium concentration in GWAS studies in Caucasians (Supplementary Data 9)[44], strengthening our findings.

We discovered four additional genes that each harbored more than five PTH concentration-associated GWAS SNPs in enhancer, DHS or super-enhancer regions. These include *APOLD1* (Apolipoprotein L Domain Containing 1, n = 5), *NDRG1* (N-Myc Downstream Regulated 1, n = 9), *RALB* (RAS Like Proto-Oncogene B, n = 64) and *SLC2A1* (Solute carrier family 2, facilitated glucose transporter member 1, n = 6). The role(s) of these genes in parathyroid biology has not been studied yet. The *RALB* gene was of particular interest, as it is a widely expressed Ras-related GTPases implicated in vesicular trafficking at the plasma

**a**

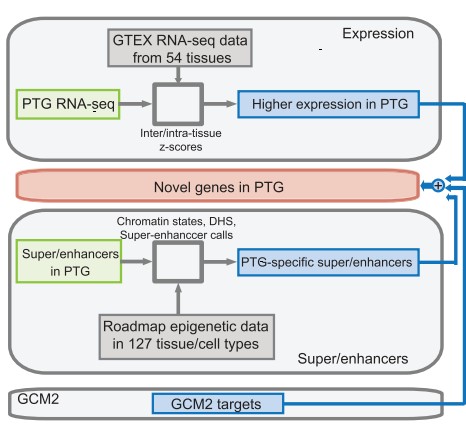

**b**

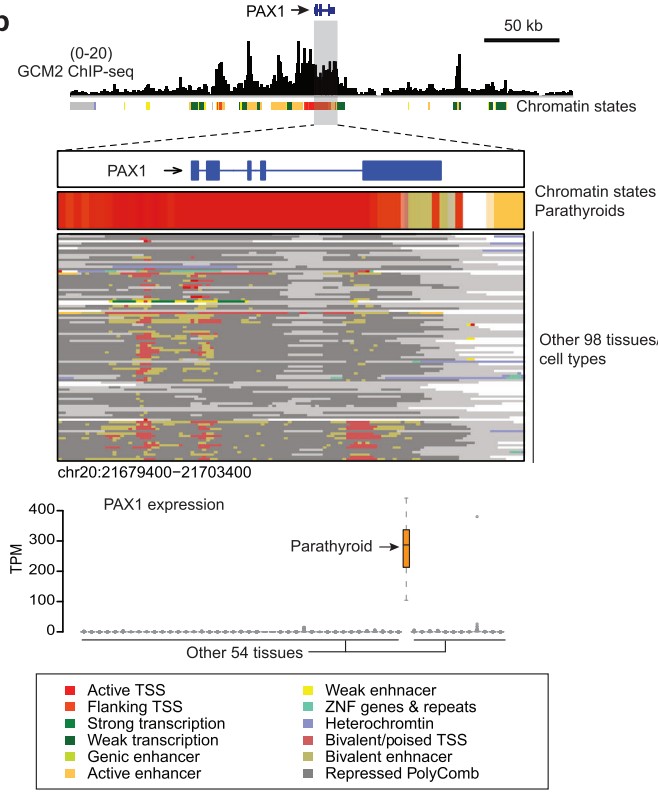

**c**

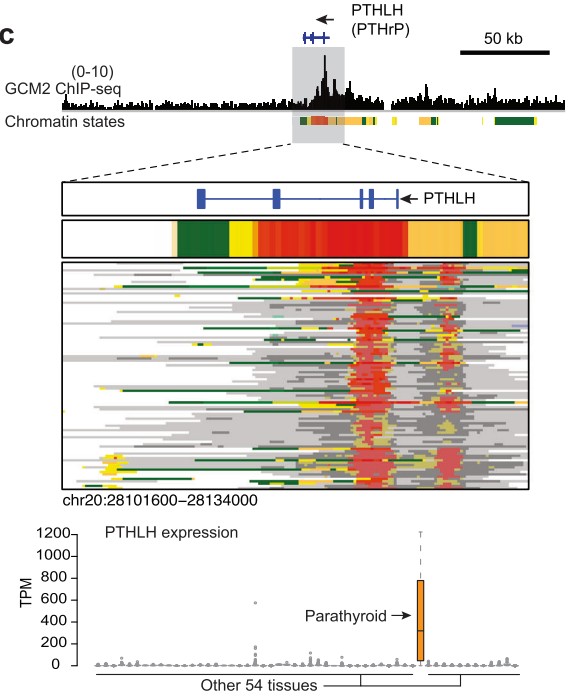

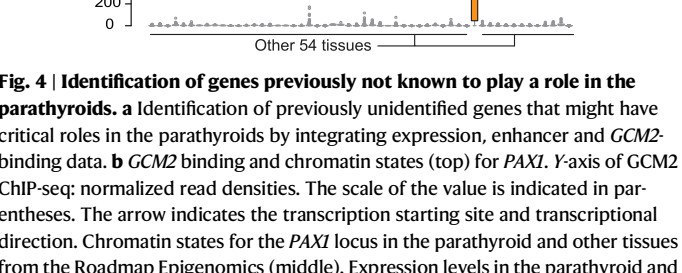

**d**

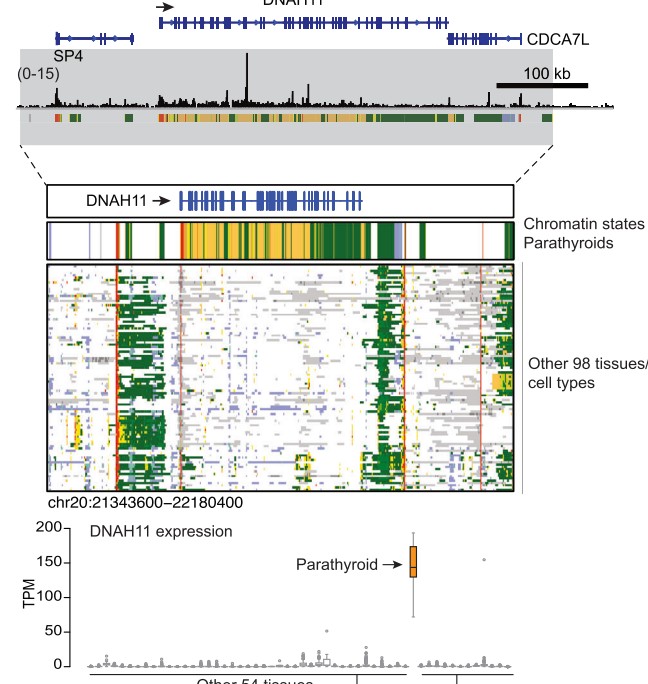

**Fig. 4 | Identification of genes previously not known to play a role in the parathyroids. a** Identification of previously unidentified genes that might have critical roles in the parathyroids by integrating expression, enhancer and *GCM2*-binding data. **b** *GCM2* binding and chromatin states (top) for *PAX1*. *Y*-axis of GCM2 ChIP-seq: normalized read densities. The scale of the value is indicated in parentheses. The arrow indicates the transcription starting site and transcriptional direction. Chromatin states for the *PAX1* locus in the parathyroid and other tissues from the Roadmap Epigenomics (middle). Expression levels in the parathyroid and other tissues from the GTEx project (bottom). Orange: parathyroid, Gray: other tissue types from the GTEx project. *n* = 8, biologically independent samples for parathyroids. The box represents the interquartile range (IQR) divided by the median, and Tukey whiskers extend to a maximum of 1.5 × IQR beyond the box. The colors in the box indicate chromatin states. **c** Same as in **b** but for *PTHLH* that is expressed in many tissue types. Broad and strong enhancers are restricted to the parathyroid. **d** Same as in **b** but for *DNAH11*.

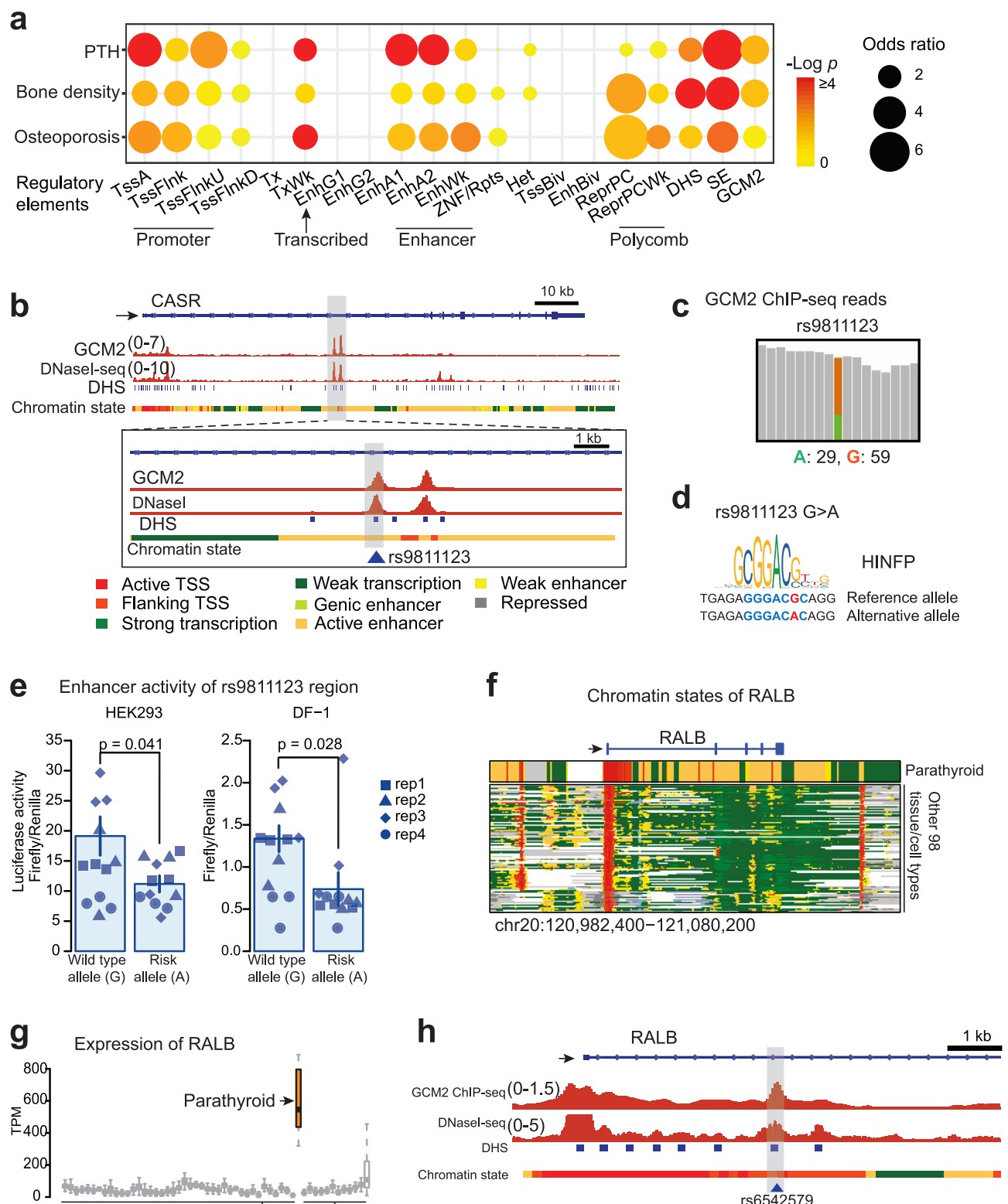

membrane[45,46]. In the neighborhood of *RALB*, we identified a parathyroid-specific super-enhancer containing a cluster of PTH-concentration-associated GWAS SNPs (Fig. 5f); Compared with 54 other tissue types, its expression level was highest in parathyroids (Fig. 5g). Specifically, rs6542579 (Supplementary Data 9) near the promoter of *RALB* co-localized with DHS and *GCM2* binding peak (Fig. 5h), and motif analysis predicted an increased affinity of the TF *DMRT2* with this SNP. Notably, the expression level of *DMRT2* itself is

the highest in parathyroids among different tissues (Supplementary Fig. 5c). Another example of PTH-linked SNP in the *MAFB* gene is presented in Supplementary Fig. 5d.

Collectively, our results demonstrated that a large fraction of GWAS SNPs associated with parathyroid hormone levels in the population were enriched in parathyroid-specific regulatory regions. For several of these SNPs, we identified supporting genomic data highlighting their functional relevance. Additionally, we experimentally

**Fig. 5 | Parathyroid-associated GWAS SNPs are enriched in regulatory regions in parathyroids. a** Enrichment of GWAS SNPs for PTH level, bone density and osteoporosis, in regulatory elements of parathyroids. *p*-values by a one-sided Fisher's exact test. The sizes of circles correspond to odds ratios and colors to *p*-values. The regulatory elements include all promoters and enhancers in parathyroids. **b** PTH level-associated SNP (rs9811123) in the parathyroid-specific intronic enhancer of *CASR*. Profiles of *GCM2* ChIP-seq, DNase-seq, chromatin states, and DHSs in parathyroids. The colors represent each chromatin state. *Y*-axis: normalized read densities for *GCM2* ChIP-seq, and DNase-seq. The scale of the value is indicated in parentheses. The PTH SNP overlaps with GCM2 peak, parathyroid-specific DHS and super-enhancer. **c** Read depths at the PTH-level SNP rs9811123 from one of the heterozygote *GCM2* ChIP-seq samples from this study showing 59 reads (67%, orange) mapping to the reference sequence while 29 (33%, green) reads mapping to the alternative sequence. **d** Motif sequence of *HINFP* around rs9811123, for which the binding affinity change by the SNP was predicted. **e** Effect of the SNP measured by luciferase assays in HEK293 and in DF-1. *n* = 4 biologically independent samples; 3 repeated measurements for each biological sample. Rep, biological

replicate. Distinct dot shapes correspond to distinct samples. *p*-values obtained through one-sided Student's *t*-tests on 4 biological replicates. The mean of 3 repeated measurements was used for each biological replicate. The bar graph displays the mean values of biological replicates with error bars of ±SEM (standard error of the mean). **f** Chromatin states around *RALB* gene in parathyroid (this study) and other tissues profiled in Roadmap Epigenomics. The same colors for chromatin states as in panel **b**, e.g., red and yellow colors indicating active promoter state and enhancer state, respectively. Arrows indicate transcription starting site and transcription direction. **g** Expression levels of *RALB* across tissue types. Orange: parathyroid, Gray: other tissue types from the GTEx project. *n* = 8, biologically independent samples for parathyroids. The box represents the interquartile range (IQR) divided by the median, and Tukey whiskers extend to a maximum of 1.5 × IQR beyond the box. **h** PTH level-associated SNP (rs6542579) in the parathyroid-specific intronic enhancer of *RALB*. Profiles of *GCM2* ChIP-seq, DNase-seq, chromatin states, and DHSs. *Y*-axis: normalized read densities for *GCM2* ChIP-seq and DNase-seq. The scale of the value is indicated in parentheses. The PTH-level SNP overlaps the *GCM2* peak, parathyroid specific DHS and super-enhancer.

validated putative enhancer regions and proposed underlying mechanisms for these GWAS SNPs.

## Discussion

In this study, we map functional elements genome-wide in human parathyroids, experimentally validate several parathyroid-specific regulatory regions, and use that information to identify key cis-regulatory elements and genes that govern parathyroid identity and function. Our exploration of the epigenetic landscape sheds light on the molecular basis of GWAS-associated loci.

We identify core regulatory circuits active in the parathyroids by using RNA-seq, histone ChIP-seq and DNase-seq and by incorporating ChIP-seq of GCM2, the parathyroid-specific transcription factor that is essential for the development and function of this organ. Our finding that *GATA3* sits on or near the top level of the parathyroid gene regulatory transcription factor circuit suggests that the dysregulation of this gene will lead to major disruption of parathyroid function, which is supported by the human disease Hypoparathyroidism, Deafness and Renal dysplasia (HDR) caused by *GATA3* mutations[22]. Our finding that *GATA3* and *MAFB* are frequent co-binding partners of *GCM2* confirmed their importance for the parathyroids. We also identified the ETS family transcription factors *ELF* or *ELK* as previously unidentified binding partners of *GCM2*.

Our approach also led to the identification of several previously unrecognized putative essential players in adult parathyroid function such as *PAX1*, *DNAH11,* and *PTHLH*. Our atlas provides the basis for future studies investigating the precise functions of these genes in this organ.

We used parathyroid adenoma in our studies instead of normal parathyroid tissue because of tissue availability, and that represents a significant limitation of the study. Parathyroid adenoma, despite a shifted calcium setpoint, exhibit all functional hallmarks of parathyroids including calcium-regulated PTH secretion; we expect that the chromatin states of parathyroid adenoma tissues would, therefore, not be significantly different from normal parathyroid tissue. However, the full extent of the similarity remains to be determined. In addition, the tissues used in our studies mainly contain parathyroid chief cells but also other cell types[47]. Single-cell approaches will be necessary to dissect the cellular heterogeneity and cell-type-specific regulation. The robust signals from our datasets and the experimental confirmation of putative enhancers indicate the high fidelity of our data for representing the parathyroid cells.

We analyzed variants identified in GWAS studies of serum concentrations of PTH and calcium, as these traits are regulated by the parathyroid glands. Indeed, genomic regions containing these GWAS SNPs were enriched for functional elements in the parathyroids and linked to genes expressed in the parathyroids. As an example, we

demonstrate that the GWAS SNP rs9811123, associated with serum PTH concentrations, is located in a parathyroid-specific enhancer bound by *GCM2* in the *CASR* gene. We detected allelic imbalance between the two alleles and showed reduced binding of the TF HINFP to the minor allele providing a mechanism for the GWAS findings.

It's worth noting that certain SNPs found to have a significant association with circulating PTH in the study by Robinson-Cohen et al.[42] were not replicated in the smaller study by Matana et al[48]. The variations in *p*-values across GWAS studies and SNPs reaching statistical significance hinge on factors such as sample size, the populations analyzed, and the methodologies used. We anticipate that certain SNPs that remain unreplicated or those with higher *p*-values may achieve significance in future studies, given more extensive variant datasets and larger sample sizes.

In conclusion, we identified functional elements genome-wide in parathyroid tissue and illustrate their potential in inferring regulatory networks and providing mechanistic insights for GWAS variants. As more parathyroid-related GWAS SNPs emerge through large databases such as the UK Biobank[49], our epigenomic data can be instrumental in uncovering more clinically relevant genomic loci in the parathyroids.

## Methods

### Sample collection
Human parathyroid adenoma tissues were collected with informed consent at Massachusetts General Hospital under protocol 2008P001466. Samples were snap frozen on dry ice and stored at −80 °C until processing.

### ATAC-seq
ATAC-seq of human parathyroid adenoma tissue was done following a previously published protocol[50] with some minor modifications. All steps of nuclei isolated were carried out on ice. 10 mg of frozen tissue was placed in a pre-chilled 2 mL Dounce with 200 μL cold 1x homogenization buffer (5 mM $CaCl_2$, 3 mM Mg(AC)$_2$, 10 mM Tris pH 7.8, 16.8 μM PMSF, 167.8 μM beta-mercaptoethanol, 320 mM sucrose, 0.1 mM EDTA, 0.1% NP40) for 5 min. 10 strokes with pestle A on ice, filtered using 100 μM MACS SmartStainers (Miltenyi Biotec, 130-098-463), then dounced 30 strokes with pestle B. 300 μL of 1x homogenization buffer was added to the sample, transferred to a micro-centrifuge tube, and centrifuged for 5 min at 500×*g* at 4 °C. Supernatant was removed, and the pellet was resuspended in a 300 μL homogenization buffer. 300 μL of 50% Iodixanol was added. 600 μL of 29% Iodixanol was added slowly under the interface without disturbing the interface and then centrifuged at 5000×*g* for 5 min at 4 °C. The supernatant was gently removed and resuspended in transposition reaction mix from the Nextera XT kit (Illumina, FC-131-1024) (25 μL 2x reaction buffer, 2.5 μL TDE1, 16.5 μL PBS, 0.5 μL 1% Digitonin (Promega,

G9441 diluted 1:1 in water), 0.5 μL Tween-20 (Roche, 11332465001), 5 μL nuclease-free water). Transposition was carried out in a thermomixer at 37 °C for 30 min with 700 rpm mixing. Reaction was purified using the Qiagen MinElute PCR purification kit, eluted in 20 μL. The transposed DNA fragments were amplified using NEBNext High-fidelity 2x PCR master mix (NEB, M0541L) for 10 cycles. Library was purified using Ampure Beads (Beckman) and sequenced at BGI Americas using DNB-seq paired-end 100 bp.

### RNA-seq

10 mg of tissue was dissected on dry ice and homogenized in 750 μL TRIzol ™ Reagent (Thermo Fisher Scientific) using 1.5 mM Zirconium Beads (Thomas Fisher Scientific, 1211U67) in TissueLyser II (Qiagen). Solution was transferred to 1.5 mL Eppendorf ™ DNA LoBind Microcentrifuge Tubes and 150 μL chloroform was added. Tubes were shaken for 30 s, incubated at room temperature for 3 min and then centrifuged at 4 °C at 12,000×$g$ for 15 min. The upper aqueous phase was transferred to a DNA LoBind tube containing 600 μL 70% ethanol, mixed by inverting, then transferred to a Qiagen RNeasy Spin column (Qiagen, 74134). Column was centrifuged for 30 s at 10,000×$g$, flow-through discarded and 700 μL of Buffer RW1 was added. Sample was centrifuged for 30 s 10,000×$g$, and rinsed twice with 500 μL Buffer RPE. Column was dried by centrifugation at 13,000×$g$ for 1 min. 30 μL of RNase-free water was added to the column and centrifuged for 1 min at 10,000×$g$. RNA quality was assessed using a bioanalyzer (Agilent Technologies) using the Agilent RNA 6000 Nano kit. Library construction and sequencing was performed by BGI Americas. The libraries were prepared using the New England Biolabs Next Ultra II DNA Library Preparation Kit designed for Illumina sequencing. Sequencing was done on the Illumina HiSeq 2000 platform, employing V3 high-output kits, and a read length of 40 base pairs (bp) was used for the sequencing process.

### ChIP-seq

Chromatin precipitation of human parathyroid adenoma tissue was performed using the EZ-Magna ChIP™ A kit (Millipore Catalog # 17-408, Billerica, MA) according to the manufacturer's instructions. Briefly, 5 mg human parathyroid adenoma tissue was minced, homogenized, and crosslinked by 1% formaldehyde for 10 min and then quenched with 0.125 mM glycine. Cells were lysed and nuclei sonicated (Bioruptor, Diagenode Inc, Denville, NJ) to achieve 200–800 bp DNA fragments. Immunoprecipitation was carried out using GCM2 (S-19) sc-79496 antibody (Santa Cruz), or normal goat IgG sc-2028 IgG (Santa Cruz). Magnet beads were used for separation and salt buffer washing. Elution of protein/DNA complexes and reverse crosslinks of protein/DNA complexes to free DNA were performed according to protocol. DNA enrichment of ChIP assay was checked by SYBR-green Real-time PCR using GAPDH as control (primers FOR: 5'-TACTAGCGGTTT TACGGGCG-3' REV: 5'-TCGAACA GGAG GAGCAGAGAGCGA-3.) The antibody information for all ChIP-seq experiments, including histone modifications, can be found in Supplementary Data 10.

### DNase-seq

DNase-Seq was performed on three biological replicates of flash-frozen human parathyroid tissue following the published protocol[51]. Briefly, nuclei are isolated from 50 mg of fresh parathyroid adenoma using Igepal CA-630 (Sigma Aldrich), followed by digestion with limiting concentrations of DNase I of an aliquot of the nuclei. Samples were treated with Proteinase K, DNA was extracted using phenol:chloroform:isoamyl alcohol and DNA fragments size-fractionated using the Invitrogen E-Gel® Agarose System. Fragments in the 175–400 bp region were collected and tested for sufficient enrichment of known DHS over non-DHS sites compared to undigested nuclear DNA. Size selection enrichment was tested by quantitative PCR using

primers (three positive control primers and three negative control primers) for constitutive DNase hypersensitive regions and insensitive regions as positive and negative controls respectively, on both the samples as well as the reserved undigested controls. Enrichment of at least 10-fold indicates successful preparation of the chromatin and DNase digest. Size selected DNA fragments underwent Illumina sequencing library preparation and quality control assessment, according to the procedures at MIT's BioMicro Center. Samples were sequenced by Illumina Hi-Seq.

### High-throughput chromosome conformation capture (Hi-C)

Hi-C was performed on frozen human parathyroid samples as described in Lieberman-Aiden et al.[52]. In brief, 10 mg of frozen sample was minced, homogenized, and fixed by 1% formaldehyde for 10 min at room temperature and then quenched with 0.125 mM glycine. Cell pellets were lysed, and DNA digested with DpnII. 5' overhangs were filled, including a biotinylated residue. The resulting blunt-end fragments were ligated. Hi-C libraries were created by selecting the biotin-containing fragments with streptavidin beads. The Hi-C libraries were sequenced on Illumina HiSeq 4000 instruments to 2 billion pair-end reads. A total of two replicates were independently generated and sequenced.

### Chip-seq and ATAC-seq processing

Reads were aligned to the hg19 genome using Bowtie version 1.2.2[53] with a unique mapping option. Other types of datasets also followed the same alignment step. We only used uniquely mappable reads for subsequent analyses. Reads from replicates were merged after checking consistency. Input normalized profiles and significant peaks were generated using MACS callpeak[41] version 1.4 with $q$-value of 0.01. *GCM2* significant peaks were further filtered with DHS sites in parathyroids.

### Chromatin state segmentation

We computed chromatin states for parathyroids using ChromHMM[15,16]. Our ChIP-seq profiles in parathyroids of H3K4me3, H3K4me1, H3K27ac, H3K36me3, H3K27me3 and H3K9me3 and the emission matrix from 18 states pre-trained in the Roadmap Epigenomics in 98 tissue/cells using the same marks[5] were used for the inputs to generate states for each segmentation (200 bp bin) in parathyroids. For Fig. 1c *GCM2* distribution at chromatin states, the background distribution was based on genome-wide frequency of each chromatin state in parathyroids. Chromatin state segmentation based on a 6 marks-model, 18 state-model, was obtained from Roadmap Epigenomics data portal (https://egg2.wustl.edu/roadmap/web_portal/).

### RNA-seq data processing

We quantified gene transcription levels in FPKM (Fragments Per Kilobase of transcript per Million mapped reads) for parathyroid samples and converted them to transcripts per million (TPM) to compare them with those in public data. For other 54 tissues, expression levels in TPM for each gene and each sample were obtained from the GTEx data portal (https://gtexportal.org/home/)[7]. tSNE was performed based on parathyroid-specific genes for which z-scores across tissues are > 7 and z-scores within parathyroids > 0 for the samples from 55 tissues including parathyroid glands. For visualization, a small amount of noise was added in Fig. 1b. To compare the expression levels of each gene in parathyroids with those in other tissues, we calculated z-scores across tissues for each gene. We reported the genes with z-scores > 7 and TPM > 50 to have transcriptional specificity in parathyroids in Supplementary Data 8.

### DNase-seq data processing

Significant peaks compared to background distribution was computed using Hotspot[54] with $q$-value of 0.01. For visualization and DHSs in

subsequent analysis, we used the data from the sample with the best quality indicated by quality metrics (e.g., Spot score) among three samples.

## Super-enhancers

We identified super-enhancers based on intensity and broadness of H3K27ac signal in parathyroids following the steps described in Whyte et al.[20]. For comparison of parathyroid super-enhancers with those from other tissues, we downloaded all available H3K27ac data (98 cell/tissues) from Roadmap Epigenomics (https://egg2.wustl.edu/roadmap/web_portal/)[5] and processed in the same way described above. To assess the enrichment of *GCM2* binding in super-enhancers, we compared the frequencies of *GCM2* within super-enhancers to those within background regions of equivalent sizes to the super-enhancers.

## TF binding prediction

We obtained de-novo motif sequences at the *GCM2* peak summits with a 100 bp margin using MEME DREAM[55]. For the de-novo motif sequences, best matched known motifs were searched using MEME TOMTOM. We used the motif sequences for the search from JASPAR[56] and Jolma et al. in 2013[57]. The motifs of GCM2 heterodimer were obtained from Jolma et al. in 2015[35]. The binding positions were predicted using MEME FIMO in Fig. 2. To predict enriched TF motifs in active parathyroid DHSs, we compared the motif prevalence in the DHSs with that in randomly generated control sequences from the background using MEME AME. The motifs from JASPAR and Jolma et al were used as above. The motifs of TFs whose expression levels were higher than 50 TPM in parathyroids were considered for the analysis.

## TF regulatory circuit

From the regulatory regions of *GCM2* overlapping with DHSs, we searched for TF motifs from JASPAR[56] or Jolma et al.[57] using MEME FIMO[55] with a *p*-value of 0.0001. Among the TFs detected, TFs that were not expressed in parathyroids were filtered out. The TFs having super-enhancers were kept for the subsequent analysis. The presence of TF motifs at the regulatory regions of the GCM2 gene was taken as evidence that these TFs exert regulatory control on GCM2. For each TF gene identified, we searched for TF motifs from DHS within a ±5 kb window from the TSS of the gene. The same procedure was done for every TFs.

## Hi-C data processing

Read alignment and contact matrices calculation after normalization were done using Juicer[58]. To ensure the quality of the Hi-C experiments, various statistics were calculated such as total aligned reads, intra- versus inter chromosomal reads, indicating a high quality of experiments. We combined the reads from biological replicates and identified significant interactions at a 5 kb resolution and FDR = 0.1 using Juicer. We obtained information on promoter-regulatory element interactions in other tissues from pcHi-C data in Jung et al.[32].

## GWAS SNP analysis

We obtained PTH-concentration GWAS SNPs with relaxed thresholds from Robinson-Cohen et al.[42]. The GWAS SNPs associated with bone density and osteoporosis were downloaded from https://www.ebi.ac.uk/gwas/[59]. The GWAS SNPs associated with facial morphology and schizophrenia were also downloaded from https://www.ebi.ac.uk/gwas/ and were used as negative control for SNPs that are not related to parathyroid functions. For the enrichment of SNPs in each chromatin feature such as chromatin states, DHSs, super-enhancers and *GCM2* peaks, *p*-values were calculated by Fisher's exact test. Read depths mapped to reference sequence and alternative sequence at SNP sites were estimated using SAMtools mpileup[60] for GCM2 ChIP-seq data.

To assess the enrichment of SNPs in super-enhancers, we compared the frequencies of SNPs within super-enhancers to those within background regions of equivalent sizes to the super-enhancers.

Enhancer potential disruption was predicted following Lee et al.[43], which is a SVM-based model where the positive training set from the DNA sequences from the regulatory regions was contrasted to the negative training set of randomized sequences with similar GC biases and repeat fractions. We trained the SVM model using DHSs in parathyroids in 300 bp overlapping with H3K27ac or H3K4me1 peaks as a positive training set. TSS-proximal sites (<2 kb from TSS) or common DHSs across tissues (i.e., DHSs in >50% of tissue types from Roadmap Epigenomics) were removed. For each SNP site, we predicted TFs to have binding affinity changes using MEME FIMO[55], HaploReg[61] and the Ensembl Variant Effect Predictor[62] and reported the results that were consistent between at least two methods.

## Enhancer activity validation

pGl4.23 (Promega) was digested with BglII and SacI, then gel purified. Enhancer regions were amplified from patient DNA using Platinum Taq (Thermo Fisher) with primers from Integrated DNA Technologies (IDT) and gel purified. Enhancer regions were digested with BglII and SacI. For CasR2, vectors and enhancers were digested with HindIII and SacI. Enhancer regions were ligated to pGl4.23 using T7 ligase (NEB) at a 3:1 molar ratio at room temperature for 30 min and then were transformed into NEB stable competent *E. coli* (high efficiency) and were spread on one Ampicillin selection plates. They were incubated overnight at 30 °C. Plasmids were extracted using Promega Wizard Plus SV Minipreps DNA purification systems. For the *CASR* enhancer SNP validation, the patient DNA used for generation of the enhancer contained both wild-type and the minor allele. Therefore, we amplified the enhancer region and cloned into the pGL4.23 and screened for the SNP using Sanger sequencing (MGH DNA Core).

HEK293 and DF1 were maintained in DMEM (Thermo Fisher Scientific) with 10% Hyclone FBS (GE Life Sciences) with 1x Penicillin–Streptomycin. For transfections, HEK293 or DF1 were plated as single cells in black 96-well plates at 10,000 cells per well; one day prior to transfection. 90 ng of enhancer plasmid and 10 ng of pRL-TK (Promega) was transfected in triplicates using Lipofectamine 2000 (Thermo Fisher Scientific). Media was changed 24 h post transfection. Enhancer activity was assayed using the Dual Luciferase Reporter Assay System.

## Statistical test

We used the Fisher exact test to assess the enrichment of *GCM2* binding in chromatin states, the enrichment of motifs at *GCM2*-binding sites, and the enrichment of GWAS SNPs in genomic regions. For enhancer activity experiments, a Student's *t*-test was used. To determine the significance of GCM2 bindings and GWAS SNPs in super-enhancers, we conducted a permutation test.

## Reporting summary

Further information on research design is available in the Nature Portfolio Reporting Summary linked to this article.

# Data availability

The sequencing data generated in this study have been deposited in the dbGaP (https://www.ncbi.nlm.nih.gov/projects/gap/cgi-bin/study.cgi?study_id=phs003302.v1.p1). The data are available under restricted access for the potential presence of information that could pose risks to human subjects. Controlled-access can be obtained upon authorization by the dbGaP Data Access Committee (DAC). The processed data generated in this study are provided in the Supplementary Information/Source Data file. The additional data used in this study are available in the GitHub (https://github.com/YLucyJung/PTG) and under accession code (https://zenodo.org/doi/10.5281/zenodo.10593362).

## Code availability

All code for the analyses in this manuscript is available in the GitHub (https://github.com/YLucyJung/PTG) and under accession code (https://zenodo.org/doi/10.5281/zenodo.10593362).

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

## Acknowledgements

This research was supported by funds from the National Human Genome Research Institute (HG006991 to B.E.B.) and the National Institute of Diabetes and Digestive and Kidney (R01DK100584 to M.M.).

## Author contributions

Y.L.J. analyzed the data with help from D.J. W.Z. performed ChIP-seq experiments of GCM2. I.L. performed ATAC-seq, RNA-seq and in vitro validation experiments. C.B.E. and B.E.B. generated ChIP-seq data of histone modifications and CTCF. S.P. consented patients and collected human surgical samples. R.S. performed DNase-seq experiments. C.R.-C. provided GWAS associated with parathyroid hormone serum concentrations. Y.H. performed Hi-C experiments. Y.L.J. and M.M. interpreted the data and wrote the manuscript. P.J.P. supervised the computational analysis. M.M. conceived and supervised the project. All authors discussed the results and commented on the manuscript.

## Competing interests

The authors declare no competing interests.
