## [Peer Review File · Nature Communications]

Epigenetic Profiling Reveals Key Genes and Cis2 Regulatory Networks Specific to Human ParathyroidsREVIEWER COMMENTS

Reviewer #1 (Remarks to the Author):

This is a comprehensive study that maps the chromatin landscape in human parathyroid tissue, identifies regulatory elements and chromatin interaction. The study defines regulatory circuits and identifies novel genes important for parathyroid gland function. The authors also demonstrated how SNPs identified through GWAS affect parathyroid regulatory elements. The authors mostly focused on two genes important for the function of parathyroid cells, CASR and PTH as well as transcription factor GCM2 important for the regulation of cell differentiation and survival.

I have several comments on the manuscript:

1. Line 82 of the Introduction section states that calcium homeostasis depends on the function of the parathyroid glands. This statement is only partially true because calcium homeostasis also depends on other parameters such as the level of the active form of vitamin D, calcitonin, osteocalcin, etc.
2. Lines 89-90 of the Introduction section, it should be stated that CASR gene expression is affected in primary and secondary hyperparathyroidism.
3. Part of the study results are generated from the human parathyroid adenoma samples. This tissue was used due to availability. Although the authors state that chromatin in parathyroid adenoma should not be significantly altered compared to normal tissue this statement has not been confirmed and therefore represents a significant limitation of the study. That should be highlighted in the text of the manuscript.
4. In the section "GWAS SNPs in parathyroid-specific enhancers" the authors state that they chose the SNPs according to the largest GWA study on circulating PTH concentrations. The study by Robinson-Cohen et al found an association of SNPs from five independent regions with serum PTH concentration. Out of five prominent SNPs only three were replicated. The same three SNPs showed the same direction of effect in the second GWA study performed by Matana et al. The SNP rs73186030 located near the CASR gene showed the lowest association ($P=4.8 \times 10^{-8}$) and was not replicated, not in the study of Robinson-Cohen et. nor in the Matana et al. study. The authors should comment on these facts in the manuscript. In addition, the authors should check the statement written in line 342 "including relaxed p-value thresholds below 5×10^{-8} ", did you mean of less significance i.e. higher than 5×10^{-8} ? Be precise with the relaxed p-value formulation.
5. However, in this paper, the authors focus on rs9811123 located in the enhancer region of the CASR gene. How was this SNP selected and what is its p-value in the GWA study? It would be useful to provide the p-value according to GWA studies for all selected and annotated SNPs in the supplementary materials. Such an incomplete explanation diminishes the value of the results.

Reviewer #2 (Remarks to the Author):

This study fills a gap in our knowledge about the epigenomic landscape of human parathyroids. The authors generated several histone marks, open chromatin maps and transcriptomes, in addition to GCM2 transcription factor ChIP-seq. The authors compare their profiles to publicly available data from a range of cell types to identify parathyroid-specific regulatory elements and then zoom into super-enhancers and genes associated with them. The authors also show the functionality of the enhancers using a luciferase assay and in Figure 5 show that known SNPs within TF motifs in these enhancers influence gene expression. Overall the paper is well written and logical.

Major comments

1. The provided GitHub link does not contain any information about the code and I could not find the raw data on dbGaP (I searched 'parathyroid' and the senior author name). Before the paper can be accepted, an identifier must be provided in the paper so that the data can be located by interested readers.

2. It is not clear which ChIP-seq antibodies were used – cat. No. and company.
3. Page 6, line 170: 433 super-enhancers were targeted by GCM, which was higher than chance. Did this calculation consider that super enhancers are longer?
4. Page 6, line 190: what does 'highly specific' mean here. How does it compare to the 'specific' as described on page 5 line 143?
5. On page 7, line 204-205, the authors argue that different functions of CASR in different cells requires different regulation of expression. Can you please provide a reference or elaborate? Is it not simply that the upstream regulators are likely different, but may regulate gene expression through a common downstream mechanism and genomic elements. In any case, this statement is used to segue into exploring enhancers, by saying the promoter is open in kidney, but the enhancers are different. Looking at Figure 2A, it looks like the promoter itself is different because the parathyroid DNaseI peak is downstream of the kidney DNaseI peak, while pancreas has a very weak peak. This also highlights the need for a reference peak as a control in the different track figures, as well as a y-axis.
6. Page 9, line 269. I understand the rationale for focusing on GCM2 only in that it streamlines the narrative. However, a step seems to be missing. For example, if the authors only generated DNaseI data first and identified parathyroid-specific open chromatin, would GCM2 be the most enriched? As the paper reads, the study ignores other regulatory regions if they do not have GCM2 binding, with even the motif enrichment only performed at GCM2 peaks, not all DNaseI peaks (Figure 3).
7. In Figure 5A we see enrichment of SNPs within regulatory regions. Is this based on all promoters and enhancers, or only the 8% and 11% that are parathyroid-specific? A negative control is necessary here – for example a gene ontology term that is completely unrelated to parathyroid function. This is especially important in light of the statement on page 11 line 388 – 'a large fraction of GWAS SNPs...'. The percentage is 23%, is this a large fraction? Relative to what? A random SNP? Or a set of SNPs associated with an unrelated phenotype?

Minor comments

1. Page 5 lines 157-161 repeats line 153-154. If chromatin states were determined using specific histone modifications, then of course those regions contain those histone marks.
2. Page 6, line 162: 'distal-promoter' region sites. What is a distal-promoter? On page 5 line 154 contains 'promoter', 'promoter-proximal' and 'distal enhancer', so distal-promoter I assume is distal enhancer?
3. There are no y-axis on any of the ChIP-seq tracks in the figures.
4. Page 10, line 305. The term 'enriched' usually implies a statistical test was performed. Is this the case for GCM2 coverage of PAX1? It is clear that there is GCM2 binding at PAX1 from the plot.
5. Page 20, line 572: more information required for the sequencing experiments. What kit? What machine, how long were the reads?

We want to express our gratitude to the reviewers for dedicating your time and effort to review our manuscript titled “Epigenetic Profiling Reveals Key Genes and Cis Regulatory Networks Specific to Human Parathyroids”. We greatly appreciate the reviewers’ supportive and insightful comments. One of the reviewers noted that ‘...This study fills a gap in our knowledge about the epigenomic landscape of human parathyroids... Overall, the paper is well-written and logical...’

We have deposited our raw data and analysis scripts in the public domain. Furthermore, we have thoroughly revised our manuscript and addressed the points made by the reviewers. Please find a point-by-point response below, along with the revised manuscript (with changes highlighted in blue), figures, and supplementary materials.

Reviewer #1

1. *“Line 82 of the Introduction section states that calcium homeostasis depends on the function of the parathyroid glands. This statement is only partially true because calcium homeostasis also depends on other parameters such as the level of the active form of vitamin D, calcitonin, osteocalcin, etc.”*

We thank the reviewer for the comment and reworded that sentence in Lines 79-81 of the revised manuscript: **Calcium homeostasis is primarily regulated by PTH and vitamin D. The parathyroid glands are the only source of PTH in the body; any disruption in their function can lead to an imbalance in calcium homeostasis.**

2. *“Lines 89-90 of the Introduction section, it should be stated that CASR gene expression is affected in primary and secondary hyperparathyroidism.”*

We agree with the reviewer and rephrased the sentence in Lines 87-88 of the revised manuscript: **the CASR gene expression is reduced in primary and secondary hyperparathyroidism, yet the underlying molecular mechanisms are not completely understood** ³⁻⁵.

3. *“Part of the study results are generated from the human parathyroid adenoma samples. This tissue was used due to availability. Although the authors state that chromatin in parathyroid adenoma should not be significantly altered compared to normal tissue this statement has not been confirmed and therefore represents a significant limitation of the study. That should be highlighted in the text of the manuscript.”*

We agree with the reviewer that this is a limitation of the study and highlighted this in Line 414 Discussion of the initial manuscript. We now expanded in Lines 433-435 of the revised manuscript: **We used parathyroid adenoma in our studies instead of normal parathyroid tissue because of tissue availability, and that represents a significant limitation of the study.**

4. *“In the section “GWAS SNPs in parathyroid-specific enhancers” the authors state that they chose the SNPs according to the largest GWA study on circulating PTH concentrations. The study by Robinson-Cohen et al found an association of SNPs from five independent regions with serum PTH concentration. Out of five prominent SNPs only three were replicated. The*

same three SNPs showed the same direction of effect in the second GWA study performed by Matana et al. The SNP rs73186030 located near the CASR gene showed the lowest association ($P=4.8 \times 10^{-8}$) and was not replicated, not in the study of Robinson-Cohen et. nor in the Matana et al. study. The authors should comment on these facts in the manuscript. In addition, the authors should check the statement written in line 342 “including relaxed p-value thresholds below 5×10^{-8} ”, did you mean of less significance i.e. higher than 5×10^{-8} ? Be precise with the relaxed p-value formulation.”

We thank the reviewer for this question and agree that certain SNPs reported in one study may not reach statistical significance in another, especially for SNPs with lower significance levels. Indeed, the two SNPs (rs73186030 and rs4443100) in Robinson-Cohen et al, that were not replicated in another cohort, had the highest p-values among the reported five SNPs, with the p-value of rs73186030 barely reaching the statistical threshold. This SNP is highly unlikely to reach the threshold in a cohort with a smaller sample size, such as the study by Matana *et al.* We have added this in Discussion Lines 453-459: **It's worth noting that certain SNPs found to have a significant association with circulating PTH in the study by Robinson-Cohen *et al.*⁴⁴ were not replicated in the smaller study by Matana *et al.*⁵¹ The variations in p-values across GWAS studies and SNPs reaching statistical significance hinge on factors such as sample size, the populations analyzed, and the methodologies used. We anticipate that certain SNPs that remain unreplicated or those with higher p-values may achieve significance in future studies, given more extensive variant datasets and larger sample sizes.**

Furthermore, we have fixed our mistake in our statement about thresholds and changed the wording to “p-value thresholds of **higher than** 5×10^{-8} ” in Line 349.

5. “... *the authors focus on rs9811123 located in the enhancer region of the CASR gene. How was this SNP selected and what is its p-value in the GWA study? It would be useful to provide the p-value according to GWA studies for all selected and annotated SNPs in the supplementary materials. Such an incomplete explanation diminishes the value of the results.*”

We have provided additional clarification regarding our focus on specific SNPs at the beginning of the paragraph: **To begin unraveling the molecular mechanism underlying GWAS SNPs associated with PTH concentrations, we focused on those SNPs located in proximity to CASR, a gene prominently involved in the regulation of PTH secretion (see preceding section).** The p-value of this SNP was 6.07×10^{-9} in the model where sex, age, and season of the PTH measurement were covariates. In addition, this SNP was also identified to have a significant association with serum calcium concentration in Caucasians in the study by O'Seaghdha *et al.* (Seaghdha *et al.* 2013), which we added in Lines 384-388.

Finally, we have provided p-values for all selected SNPs in our study in **Supplementary Table 9.**

Reviewer #2

Major comments:

1. “*The provided GitHub link does not contain any information about the code and I could not*

find the raw data on dbGaP (I searched 'parathyroid' and the senior author name). Before the paper can be accepted, an identifier must be provided in the paper so that the data can be located by interested readers."

Thank you for raising this important issue. We have now deposited our data to **dbGaP** (https://www.ncbi.nlm.nih.gov/projects/gap/cgi-bin/study.cgi?study_id=phs003302.v1.p1) and codes to **github** (<https://github.com/YLucyJung/PTG>).

2. *"It is not clear which ChIP-seq antibodies were used – cat. No. and company."*

We used the GCM2 (S-19): sc-79496 (Santa Cruz Biotechnology, Inc) for GCM2 ChIP-seq. We have added this in the Method section and summarized the antibody information for all ChIP-seq experiments, including histone modifications in **Supplementary Table 10**.

3. *"Page 6, line 170: 433 super-enhancers were targeted by GCM, which was higher than chance. Did this calculation consider that super enhancers are longer?"*

The size of the region was considered in our calculation. We have added this information in the **Methods**, Super-enhancer section.

4. *"Page 6, line 190: what does 'highly specific' mean here?..."*

We initially intended "highly specific" to mean "present in at most one other tissue type" as in parenthesis of the original sentence. We agree with the reviewer that this is unclear. Now we rephrased it to **"We identified 138 genes with super-enhancers that displayed specificity to parathyroids, appearing in no more than one additional tissue type"**.

5. *"On page 7, line 204-205, the authors argue that different functions of CASR in different cells requires different regulation of expression. Can you please provide a reference of elaborate? Is it not simply that the upstream regulators are likely different, but may regulate gene expression through a common downstream mechanism and genomic elements. In any case, this statement is used to segue into exploring enhancers, by saying the promoter is open in kidney, but the enhancers are different. Looking at Figure 2A, it looks like the promoter itself is different because the parathyroid DNaseI peak is downstream of the kidney DNaseI peak, while pancreas has a very weak peak. This also highlights the need for a reference peak as a control in the different track figures, as well as a y-axis. "*

We thank the reviewer for these important points and agree with them. We have toned down the statement of "different functions of CASR" and changed it into "different expression levels" and provided more references. We edited the sentence on line 205 which now reads: **The expression levels of the CASR differ between tissues (GTEx Consortium 2020; Uhlén et al. 2015)(Extended Fig. 2a). This variation is likely due to differences in enhancer utilization (Ong and Corces 2011; Ko et al. 2017).**

We also agree with the reviewer's point that there are variations in promoters and near promoter-proximal regions as well as distal regions in the open chromatin profiles. Now the sentence reads (line 205 of the revised manuscript): **There were striking differences in the usage of both promoter-proximal and distal enhancers.**

Furthermore, we have added scales in the y-axis with descriptions in the figure legend as well as putting a reference peak at the promoter of CCDC58 that is universally expressed with similar expression levels across tissues in **Extended Fig. 2b**.

6. “Page 9, line 269. I understand the rationale for focusing on GCM2 only in that it streamlines the narrative. However, a step seems to be missing. For example, if the authors only generated DNaseI data first and identified parathyroid-specific open chromatin, would GCM2 be the most enriched? As the paper reads, the study ignores other regulatory regions if they do not have GCM2 binding, with even the motif enrichment only performed at GCM2 peaks, not all DNaseI peak...”

We thank the reviewer for these comments. We have performed additional analysis and added another step on the identification of enriched TFs in parathyroid DHSs before focusing on GCM2, by adding the description below and providing the results in **Supplementary Table 7: Our initial step involved the identification of TFs that are enriched within open chromatin regions in parathyroid tissue. By conducting motif analysis, we identified a total of 26 TFs (Supplementary Table 7; Methods) with GCM2 ranking among the top 4. Due to the unique expression of GCM2 in parathyroids and the high prevalence of its motif in DHSs, we further explored the regulatory network among these TFs by leveraging GCM2 ChIP-seq data. This allowed us to uncover a network of TFs centered around GCM2.**

7. “In Figure 5A we see enrichment of SNPs within regulatory regions. Is this based on all promoters and enhancers, or ...are parathyroid-specific? A negative control is necessary here – for example a gene ontology term that is completely unrelated to parathyroid function. This is especially important in light of the statement on page 11 line 388 – ‘a large fraction of GWAS SNPs...’. The percentage is 23%, is this a large fraction? Relative to what? A random SNP? Or a set of SNPs associated with an unrelated phenotype?”

We thank the reviewer for these important comments. Figure 5A shows the results from all promoters and enhancers in parathyroids. We clarify this in the **figure legend**. 23% was a significantly large fraction ($p < 0.001$, a permutation test, Extended Figure 5A), compared to those within background regions of equivalent sizes to the super-enhancers. Details are added to **Methods** section.

In addition, we added **negative controls** by assessing the enrichment of SNPs associated with phenotypes unrelated to parathyroid function. The enrichment analysis for **SNPs associated with facial morphology and schizophrenia** (which are unlikely to be related to parathyroid function) showed different patterns from those with parathyroid-related phenotypes—they were not enriched for active enhancers, but for weak Polycomb repressive regions. These results are now included in **Extended Fig. 5b**.

Minor comments”

1.” Page 5 lines 157-161 repeats line 153-154. If chromatin states were determined using specific histone modifications, then of course those regions contain those histone marks.”

We have shortened the redundant description in Lines 157-161.

2. *“Page 6, line 162: ‘distal-promoter’ region sites. What is a distal-promoter? On page 5 line 154 contains ‘promoter’, ‘promoter-proximal’ and ‘distal enhancer’, so distal-promoter I assume is distal enhancer?”*

Although the term is frequently used in the genomics field, we have replaced the term to “distal enhancer” to avoid any confusion.

3. *“There are no y-axis on any of the ChIP-seq tracks in the figures.”*

We have added scales in the y-axis for all genome browser track plots and descriptions for the y-axis in the figure legends for Figures 2-5.

4. *“Page 10, line 305. The term ‘enriched’ usually implies a statistical test was performed. Is this the case for GCM2 coverage of PAX1? It is clear that there is GCM2 binding at PAX1 from the plot.”*

Yes, GCM2 reads were significantly enriched compared to input reads in terms of Poisson distribution p -value $< 1e-5$. We have added this in Line 310.

5. *“Page 20, line 572: more information required for the sequencing experiments. What kit? What machine, how long were the reads?”*

The libraries were prepared using the New England Biolabs Next Ultra II DNA Library Preparation Kit designed for Illumina sequencing. Sequencing was done on the Illumina HiSeq 2000 platform, employing V3 high-output kits, and a read length of 40 base pairs (bp) was used for the sequencing process. This information was updated to the **Methods**.

REVIEWERS' COMMENTS

Reviewer #2 (Remarks to the Author):

I thank the authors for addressing all my comments, which simply strengthens their already solid manuscript.

Thank you for the reviewers' feedback. The reviewers found our responses in the previous revision to be satisfactory and had no further concerns.